# Air Pollution-Related Respiratory Diseases and Associated Environmental Factors in Chiang Mai, Thailand, in 2011–2020

**DOI:** 10.3390/tropicalmed7110341

**Published:** 2022-10-31

**Authors:** Chalita Jainonthee, Ying-Lin Wang, Colin W. K. Chen, Karuna Jainontee

**Affiliations:** 1Veterinary Public Health and Food Safety Centre for Asia Pacific (VPHCAP), Faculty of Veterinary Medicine, Chiang Mai University, Chiang Mai 50100, Thailand; 2Center of Excellence in Veterinary Public Health, Faculty of Veterinary Medicine, Chiang Mai University, Chiang Mai 50100, Thailand; 3School of Public Health, Taipei Medical University, Taipei 11031, Taiwan; 4Graduate Institute of Environmental Engineering, National Taiwan University, Taipei 10617, Taiwan; 5Southeast Bangkok College, Bangkok 10260, Thailand; 6Sustainable Management Association, Bangkok 10230, Thailand; 7Faculty of Science and Agricultural Technology, Rajamangala University of Technology Lanna, Chiang Rai 57120, Thailand

**Keywords:** air pollution, environmental factor, health risk, influenza, particulate matter, PM_2.5_, pneumonia, Thailand

## Abstract

The unfavorable effects of global climate change, which are mostly the result of human activities, have had a particularly negative effect on human health and the planet’s ecosystems. This study attempted to determine the seasonality and association of air pollution, in addition to climate conditions, with two respiratory infections, influenza and pneumonia, in Chiang Mai, Thailand, which has been considered the most polluted city on Earth during the hot season. We used a seasonal-trend decomposition procedure based on loess regression (STL) and a seasonal cycle subseries (SCS) plot to determine the seasonality of the two diseases. In addition, multivariable negative binomial regression (NBR) models were used to assess the association between the diseases and environmental variables (temperature, precipitation, relative humidity, PM_2.5_, and PM_10_). The data revealed that influenza had a clear seasonal pattern during the cold months of January and February, whereas the incidence of pneumonia showed a weak seasonal pattern. In terms of forecasting, the preceding month’s PM_2.5_ and temperature (lag1) had a significant association with influenza incidence, while the previous month’s temperature and relative humidity influenced pneumonia. Using air pollutants as an indication of respiratory disease, our models indicated that PM_2.5_ lag1 was correlated with the incidence of influenza, but not pneumonia. However, there was a linear association between PM_10_ and both diseases. This research will help in allocating clinical and public health resources in response to potential environmental changes and forecasting the future dynamics of influenza and pneumonia in the region due to air pollution.

## 1. Introduction

Climate change and air pollution are primarily the consequences of human activities and have been worldwide issues for decades. Exposure to air pollutants is connected with significant long-term unfavorable health impacts and premature mortality, particularly respiratory and cardiovascular illnesses [1,2,3,4,5,6,7]. Among air pollutants, such as ozone (O_3_), nitrogen dioxide (NO_2_), sulfur dioxide (SO_2_), carbon monoxide (CO), and particulate matter (PM) with aerodynamic diameters of less than or equal to 10 µm (PM_10_) or 2.5 µm (PM_2.5_), research on the latter two has become increasingly focused on the fact that they are small enough to travel long distances over time and can become deposited in the lower airways of humans and animals, thereby posing a greater health risk [2,4]. Seasonal analyses and the link between climatic factors and human diseases have been the focus of a number of research studies [8,9,10,11,12,13,14,15], while studies of PMs have increased due to its burden on human respiratory health [16,17,18,19,20]. PM_2.5_ and PM_10_ have been recognized as indicators of air quality and have been linked to human respiratory illnesses. 

Thailand has established an extensive air pollutant monitoring network over a number of decades with the aim of providing updated empirical data on ambient air pollutants [21,22]. Since 2018, with the growing concern about the health burdens associated with air pollutants, the Ministry of Public Health has been monitoring the health consequences of air pollution and PM_2.5_ in accordance with Thailand’s national surveillance of human diseases [23]. Respiratory diseases (namely, chronic obstructive pulmonary disease, asthma, pneumonia, influenza, acute pharyngitis, chronic rhinitis, bronchitis, and lung cancer), cardiovascular diseases (namely, ischemic heart disease and cerebrovascular disease), conjunctivitis, and dermatitis are included on the list of diseases monitored for air pollutant-related health effects. Nonetheless, surveillance of air pollutant-related diseases has been carried out mostly in Bangkok and the surrounding areas. Influenza and pneumonia are the only diseases that have been continuously monitored in all provinces across the country and reported in the national epidemiological surveillance system for several decades. 

Chiang Mai, a province in the northern region with intermittently high levels of ambient air pollution, was reported in 2019 to have the maximum 24-h average PM_2.5_ concentration of 228 µg/m^3^ and an annual average of 36 µg/m^3^, which exceeded the national standard of 50 µg/m^3^ and 25 µg/m^3^, respectively [22]. Although reports of the health consequences of air pollutants are on the rise in Thailand, regulations and interventions are not being implemented effectively to solve the concerns. Since 2014, Chiang Mai has been listed in the top five cities for influenza cases, and occasionally in the top ten for pneumonia cases. There is limited data available for air pollutant-related disease surveillance and reporting, which is considered a health information gap, given that high PM concentrations have been reported and classified as hazardous to human health for half of the year in the province with the highest population in the region.

Our research question was, in addition to climate factors, whether there is an association between the diseases of interest and air pollutants. Our objective was to use influenza and pneumonia as indicators of the Chiang Mai population’s respiratory health in relation to air pollution. We investigated seasonality of the diseases as well as the associations between the diseases and environmental variables (temperature, precipitation, relative humidity, PM_2.5_, and PM_10_) from 2011–2020. The findings from the study will benefit policymakers, public health authorities, and stakeholders in order to develop proper intervention and response plans for improving the quality of life of people in the area.

## 2. Materials and Methods

### 2.1. Study Area

This study was carried out in Chiang Mai province, which is located in northern Thailand (coordinates: 18.796143, 98.979263) with an altitude of 310 meters above sea level. Chiang Mai has a total population of 1.78 million (in 2020) and a population density of 89 people/km^2^ [24,25]. It is the second largest province of Thailand, secondary to Nakhon Ratchasima province, which covers an area of 20,107 km^2^. It is bordered by Chiang Rai, Lamphun, and Lampang to the east, Tak to the south, Mae Hong Son to the west, and Shan State of Burma to the north [26]. Chiang Mai is 696 km north of Bangkok. The general area of Chiang Mai is mountainous with brushwood. There is a plain in the middle along both sides of the Ping River, and it contains the highest mountain in Thailand, Doi Inthanon, with an altitude of 2,565 meters above sea level. The condition of the area is divided into two types, namely, the mountainous area, mainly in the north and the west of the province, accounting for about 80% of the province’s area, and the watershed plains and the foothill plains, which are scattered between the valley extending in the north–south direction. Chiang Mai is a province with a relatively cool climate most of the year. The monthly average temperature in 2020 was 27.6 °C, the average rainfall was 120.6 mm, and it has an average relative humidity of 65%. Chiang Mai’s climate is under the influence of two types of monsoons, namely, the southwest monsoon and the northeast monsoon [26]. The climate can be divided into three seasons: the rainy season starts from mid-May and lasts until mid-October, the cool season starts from mid-October and lasts until mid-February, and summer starts from mid-February and lasts until mid-May.

### 2.2. Data Collection

The number of influenza and pneumonia cases was obtained from the *Annual Epidemiological Surveillance Report* (AESR), Bureau of Epidemiology, Department of Disease Control, Ministry of Public Health of Thailand, from 2011 to 2020 [27]. Influenza and pneumonia are diseases classified in a group of infectious respiratory diseases. The case definition of influenza, defined by the AESR, is an acute respiratory viral infection with important clinical symptoms of a sudden high fever, runny nose, sore throat, coughing, headache, muscle aches, and fatigue. A causative agent of the disease is the influenza virus, which is composed of four types: A, B, C, and D. The incubation period of the disease ranges from 1 to 3 days. When using solely clinical signs, diagnosing suspected influenza patients is difficult. Laboratory tests are undertaken to confirm the disease. Pneumonia is an acute respiratory infection and was reported as the primary cause of death in those less than 5 years of age, caused by viruses, bacteria, or fungi. There are two types of pneumonia based on predisposing causes: infectious and non-infectious, with the prior type causing more severe impacts on human health. Common symptoms of pneumonia include fever, coughing, chest retraction, nasal flaring, or other signs of cardiac failure. The incubation period of pneumonia can be short, at less than 3 days, or longer than 4 weeks. Diagnosis of pneumonia is based on clinical signs of fever, coughing, and difficulty with breathing in conjunction with crepitation and bronchial breath sounds. In addition, chest X-rays and laboratory tests are also used for confirmation of the disease.

We extracted the monthly cases of influenza and pneumonia from 2011 to 2020 from the AESR, using the clinical diagnosis coding ICD-10, with regard to the principal diagnosis of influenza (J10, J11) and pneumonia (J12–J18). The number of reported cases was acquired from a hospital-based surveillance system of a provincial public health office that compiled cases reported from government hospitals, sub-district health promoting hospitals, and some private hospitals. The case reports from the local level were then submitted to the provincial office and to the Bureau of Epidemiology consecutively. At the Bureau of Epidemiology, the provincial data were then validated before publication in the *Weekly Epidemiological Surveillance Report* (WESR) and the AESR via the following website: https://ddc.moph.go.th/doe, accessed on 24 June 2022.

Additionally, data on the mid-year population of Chiang Mai were also obtained from the Bureau of Epidemiology and were used to calculate the monthly incidence rate per 100,000 population using the following Equation (1):(1)$$\begin{array}{c} monthly\;incidence\;rate\\ per\;100,000\;population \end{array}=\left[\frac{no.\;of\;cases\;in\;that\;month}{no.\;of\;population\;in\;that\;year}\right]\times 100,000$$

It was assumed that the human population was constant on a yearly basis during the study period, with a 0.9% yearly average growth.

Furthermore, the cases of influenza and pneumonia related to air pollution that were directly reported to the PM_2.5_ health data center (HDC) since July 2020 were obtained from the database (https://hdcservice.moph.go.th, accessed on 24 June 2022).

In parallel, monthly meteorological data, including the average temperature (°C), average relative humidity (%), and total precipitation (mm), for the same period were obtained from the Thai Meteorological Department from one of two weather stations in Chiang Mai: station#327501, coordinates: 18.771667, 98.969167, located in Muang district (Figure 1). The meteorological data from station#327202 (coordinates: 19.925833, 99.048333), located in a remote mountain area in Fang district, with an altitude of 1,400 meters above sea level, had significantly different weather data values that did not represent the actual weather of Chiang Mai (Appendix A). Thus, only weather data from station#327501, located in the Muang district, were used in this study. Additionally, the monthly average PM_2.5_ and PM_10_ data were obtained from two stations (station#35T, coordinates: 18.837148, 98.970829, and station#36T, coordinates: 18.790918, 98.988023, both located in the Muang district) from the Pollution Control Department. All the incidence and environmental data were then placed into Microsoft Excel spreadsheets for further analysis. The distributions of the monthly incidence of influenza and pneumonia, climate data, and the concentration of air pollutants are shown in Appendix A.

### 2.3. Data Analyses

The monthly incidence rates (per 100,000) were calculated during the study period. A seasonal-trend decomposition using loess (STL) was used to evaluate the seasonality of influenza [28]. This method decomposes a time-series dataset into three parts: trend, seasonal, and remainder components on a 12-month basis. An equation depicting an additive decomposition is shown as Equation (2): (2)$$y_{t\;}=S_{t}+T_{t}+R_{t}\;$$
where *y_t_* is the data, *S_t_* is the seasonal component, *T_t_* is the trend-cycle component, and *R_t_* is the remainder component, all at period *t*.

In addition, a seasonal cycle subseries (SCS) plot and an unconditional negative binomial regression (NBR) model were used to evaluate the monthly variability with a statistical analysis of the incidence rate ratio (IRR) with a 95% confidence interval (CI). Overdispersion (a variance larger than the mean) was observed in the monthly case counts, suggesting that the NBR models with an overdispersion term (alpha, α) were preferable to the Poisson models. For the analysis, the “*nbreg*” STATA command with the dispersion of the mean and “*irr”* option was utilized. Consequently, the IRRs were calculated. In order to evaluate the monthly differences, univariate NBR models were constructed where May was used as a reference month for influenza and June was used as a reference for pneumonia.

For the variable screening, correlations among all the predictor variables were observed. The monthly climate data (temperature, precipitation, and relative humidity) and pollutant concentration data (PM_2.5_ and PM_10_) were analyzed using Spearman’s correlation (*r_s_*). Correlations among all the predictors were investigated and the preceding month’s values were also considered in the models. If the correlation coefficient was greater than 0.7 or less than −0.7, showing a strong association between the predictors [29], one of them was omitted from the model. Furthermore, if there was no evidence of linearity, a quadratic term of the predictors was considered in the model. Several multivariable NBR models were used to assess the association between the diseases of interest and environmental variables (temperature, precipitation, relative humidity, PM_2.5_, and PM_10_) for evidence of overdispersion. In addition, the correlations between the original monthly variables and their lag1 (the preceding month) were explored in order to assess the potential lagged effect in the model. Variables with a *p*-value of *<* 0.05 were considered significant in the final models. Either an original variable or a quadratic term was significant in the model; both variables were forced into the model. To account for the unmeasured yearly predictor, a random year effect was added using the “*vce(cluster)*” option in the “*nbreg*” command to adjust the standard error for 10 clusters.

All the data were recorded in Microsoft Excel and analyzed using RStudio, version 2021.09.1 (RStudio, PBC, Boston, MA, USA), and STATA, version 14.0 (StataCorp LLC, College Station, TX, USA). A figure of the Spearman’s correlation results was constructed using GraphPad Prism, version 9.4.1 (GraphPad Software, San Diego, CA, USA). The maps were generated using QGIS, version 3.26.0 (Quantum GIS development Team, Boston, MA, USA).

## 3. Results

### 3.1. General Information of Climate and Air Pollutants

Table 1 displays the ten-year average monthly meteorological and air pollutant data from 2011 to 2020. The average temperature in Chiang Mai ranged from 22.9 °C during the dry and cool season to 30 °C during the hot season. The relative humidity ranged from 53.5% to 79.6%, and the total precipitation ranged from 7.8 mm to 220.9 mm. In addition, a report on the air pollutants revealed that the monthly PM_2.5_ concentration ranged from 14.2 µg/m^3^ to 80.9 µg/m^3^ and that the PM_10_ concentration ranged from 24.9 µg/m^3^ to 101.7 µg/m^3^.

### 3.2. Incidence Rates and Seasonal Decomposition of Time Series of Influenza and Pneumonia

Between 1 January 2011 and 31 December 2020, 84,075 cases of influenza were reported to the system. The influenza incidence rate was the highest in 2019 (1278.8 per 100,000 population; 95%CI = 669.9, 1887.7) and the lowest in 2013 (161.6 per 100,000 population; 95%CI = 48.3, 274.8). (Appendix A). In most years, the monthly incidence rates (per 100,000 population) exhibited cyclical peaks between January and February, with substantially higher incidence rates between October 2018 and March 2020 (Appendix A). Additionally, 90,239 cases of pneumonia were recorded. The incidence rate of pneumonia was the highest in 2018 (697.8 per 100,000 population; 95%CI = 629.8, 765.7) and the lowest in 2020 (404.6 per 100,000 population; 95%CI = 289.5, 519.6) (Appendix A). In most years, the monthly incidence rates (per 100,000 population) exhibited cyclical peaks in January to February and another peak in September, with substantially higher incidence rates between May and December 2016 (Appendix A).

The STL plot shows a seasonal pattern of influenza (Figure 2A) with a peak in the first few months of the calendar year (January–February) and a gradual decline, followed by a smaller second peak of the incidence rate in the third quarter of the year. The trend plot indicated an increasing trend of influenza incidence over the period of observation between 2011 and 2019. After the second half of 2019, this trend sharply decreased until the end of the period of the study. The remainder component showed varying residuals with observed large values over the past few years. In contrast, the trend plot of pneumonia showed an increasing trend with two cycles of peaks in 2012 and 2018, followed by a declining trend to the lowest point at the end of the observation (Figure 2B). The seasonal pattern of pneumonia showed annual peaks in the first few months of the year and near the end of the third quarter. The remainder component showed varying residuals with intermittently large values.

The SCS plot (Figure 3) confirmed the patterns of influenza noted above and showed that the highest incidence rate was in February, whereas the lowest was in May. An unconditional NBR model showed that, compared with May, there was a significantly higher incidence of influenza starting from July through March, whereas April and June were not statistically significantly different from the reference month (Table 2). Additionally, the SCS plot of pneumonia showed the highest incidence in September and the lowest incidence in June. There were significantly higher incidence rates of pneumonia from January to March and from August to November, when June was used as a reference (Table 2).

Considering the monitoring of air-pollution-related illnesses, which are directly reported to the HDC PM_2.5_ database of the Bureau of Epidemiology since July 2020, the proportions of influenza and pneumonia cases directly attributable to air pollution were 75% and 70%, respectively, of the number of cases reported to the AESR during the same period. The San Pa Tong, Hod, and Mae Taeng districts reported a high number of cumulative influenza cases related to air pollution during the available time period, while the Kalayaniwattana, Wiang Haeng, and Mae On districts reported few or zero cases (Figure 4). In addition, cumulative pneumonia cases were more prevalent in the districts of Mae Rim, Hang Dong, and Muang than in Mae On, Kalayaniwattana, and Doi Lo. 

### 3.3. Association between Influenza and Pneumonia Incidences and Environmental Factors

As an overview, the temperature in Chiang Mai ranged from 21.0 °C to 32.4 °C, with an average of 27.0 °C. The relative humidity was between 45% and 82% (average 69.4%). The total precipitation (rainfall) ranged between 0 mm and 411 mm with an average of 93.5 mm. Regarding the air pollutant measurements, the lowest monthly concentrations of PM_2.5_ and PM_10_ were 9.1 µg/m^3^ and 18.5 µg/m^3^, respectively, whereas the highest concentrations were 110 µg/m^3^ and 124 µg/m^3^, respectively, with the monthly median concentrations of PM_2.5_ equal to 22 µg/m^3^ and of PM_10_ equal to 35.8 µg/m^3^, as shown in Table 1.

Due to a very strong correlation between PM_2.5_ and PM_10_ (*r_s_* = 0.89) (Figure 5), the NBR of a PM_2.5_ variable was tested only with the PM_10_ of the preceding month (lag1), and vice versa. We separately developed models for relative humidity and the PM variables (PM_2.5_ and PM_10_) due to collinearity. Additionally, the humidity variables (the original variable and its lag) were excluded from the models that contained PM_2.5_ lag1 and PM_10_ lag1 due to the abovementioned reason. For influenza, the NBR models were developed, and three significant models were selected, as follows: Model 1 composed of the PM_2.5_ of the preceding month and its quadratic term. This model predicted that the influenza incidence rate would gradually decline as the monthly PM_2.5_ of the preceding month rose to its median value of 22 µg/m^3^ but that it would dramatically increase as the PM_2.5_ concentration rose past 22 µg/m^3^ (Table 3). Model 2 showed that a 10 µg/m^3^ increase in the PM_10_ concentration in the same month corresponded to a 1.2% increase in the influenza incidence rate. Lastly, Model 3 predicted the incidence rate of influenza based on the temperature data of the preceding month, predicting a 21.9% decrease in the current month’s incidence rate when there was a temperature increase of 1 °C. According to our analysis, precipitation lag1, relative humidity, and PM_2.5_ in the current month were not significant in the models.

Additionally, we developed and selected three models regarding the association between the pneumonia incidence rate and the environmental variables. Model 4 showed that a 10 µg/m^3^ increase in the PM_10_ concentration corresponded to a 0.2% increase in the influenza incidence rate (Table 3). Model 5 showed that a 1 °C increase in the temperature of the same month corresponded to a 3.3% decrease in the pneumonia incidence rate. Lastly, Model 6 displayed that a 1 °C increase in the temperature of the preceding month corresponded to a 4.9% decrease in pneumonia incidence while holding the relative humidity lag1 variable in the model constant. If the relative humidity of the preceding month was increased by 1%, the incidence of pneumonia in the current month would be expected to increase by 0.7%, holding the temperature lag1 variable in the model constant. However, from the analysis, we found that relative humidity, precipitation and its lag, PM_2.5_ and its lag, and PM_10_ lag1 were not significant in the models.

## 4. Discussion

### 4.1. Influenza and Associated Environmental Factors

This study was conducted to find an association between environmental factors and respiratory diseases in Chiang Mai. Regardless of age and gender, monthly health data were used for the analysis. According to the 2011–2020 study period, we discovered that the seasonality of influenza, with a peak in the dry and cool season during January and February, corresponded to the winter months [30]. Due to the fact that the cases reported in the AESR were non-type identified, an event-based surveillance of influenza in 2020 found that influenza A was the most prevalent type and the primary type, resulting in mortality in Thailand [31]. According to Somayaji’s study, the highest rate of hospitalization was discovered in winter, which corresponds to the period of influenza circulation [8]. Multiple models with different combinations of variables were analyzed to find the association between influenza and associated environmental factors, and the final models were chosen primarily based on the lag1, which was taken from the preceding month’s variables due to the predictability of the current month. Furthermore, additional models in which the variables had a linear relationship with the diseases of interest were investigated. From the models, temperature (data not shown) and PM_10_ showed a linear association with influenza incidence. An increasing PM_10_ concentration corresponded to an increased number of influenza cases, while temperature had a negative effect on influenza infections. In addition, in terms of forecasting, we found that temperature, precipitation (data not shown), PM_2.5_, and PM_10_ (data not shown) of the preceding month were significant in the models. 

It is obvious that meteorological factors influence influenza infections and transmission. However, the relationship between air pollution and the disease is less understood. A study conducted in Beijing on the correlation between influenza-like illness (ILI) and PM_2.5_ indicated a high positive association between ILI risk and PM_2.5_ during the flu season (October to April) but no significant association during the non-flu season (May to September) [32]. Numerous viruses would utilize atmospheric PM as a vector or carrier [33]. Both coarse (>5 μm) and fine (≤5 μm) aerosols exhaled from infected individuals have been reported to contain influenza virus RNA, with higher viral RNA copy numbers detected in fine aerosols, which has been described as airborne transmission [34,35,36]. PMs smaller than 10 μm are small enough to float in the air for extended durations, and influenza viruses could adhere to the surface of these particles by coagulation to become droplet nuclei or aerosols, promoting the spread of the viruses through the air and making individuals susceptible to infection via inhalation or physical contact [37,38,39]. High concentrations of PMs resuspended in the air encourage the virus to bind to the particles, hence promoting virus transmission [32]. Almost 80% of inhaled PM_2.5_ can reach the upper-to-lower airway, and 47% can reach the alveolar epithelium for influenza virus infection, according to the findings of Hsiao’s study [39]. Consequently, PM_2.5_ may provide a suitable way for shedding and transporting influenza viruses during atmospheric transportation [39].

In addition, several alternative explanations were proposed, including the idea that PMs cause a transient immunosuppressive pulmonary microenvironment by damaging the airway epithelial cells and disrupting the barrier between the airway and the rest of the lung [37,40]. Interestingly, in contrast to our hypothesis, another study in Beijing revealed a correlation between air pollution and outpatient visits for ILI during both non-outbreak and outbreak seasons. Using disposable face masks and altering outdoor activities may reduce outpatient visits during an outbreak; thus, the authors concluded that air pollutants may be associated with a decreased risk of outpatient visits, while an increased risk was observed in the non-outbreak ILI period [41]. In Liu’s study, data on atmospheric visibility, which relates to ambient air quality, were used to determine an association with influenza. There was a non-linear association between atmospheric visibility and influenza, with low visibility indicating a high concentration of air pollutants and influenza cases, showing a positive relationship [42]. Regarding the climate zone, tropical and subtropical regions have been mentioned as having an influenza seasonality that is less prominent than that in temperate regions due to their diverse seasonal patterns [43,44]. Despite Thailand’s location in a tropical region, our study’s SCS plot revealed a definite peak in January and February, corresponding to the cold winter months in other Northern Hemisphere countries [45]. Environmental variables have been discovered to improve influenza forecasts and may play a role in influenza illness in a number of different ways [8]. The experiments conducted in Lowen’s research demonstrated that temperature and humidity play a role in influenza viral transmission. Low relative humidity, caused by indoor heating and the cold temperatures of winter, facilitates the propagation of the influenza virus [46]. In addition to climatic reasons, explanations for influenza seasonality include oscillations in host immunological competence mediated by seasonal factors, such as melatonin and vitamin D levels, and seasonal changes in host behavior, such as gathering indoors during adverse weather [9,10]. Inverse associations between influenza transmissions and temperature and relative/absolute humidity have also been reported in studies from throughout the world [11,12,47,48,49]. However, in this study, we did not find an association between the incidence of influenza and relative humidity. Thus, this factor was excluded from our model. For the variable of PM_2.5_ lag1, for which linearity was not observed, a quadratic term of the variables was considered and added to the models. The influenza incidence of the current month gradually declined when the lag1 data of PM_2.5_ were below their means (as shown in Table 1); above these levels, the incidence rate began to increase, producing a U-shaped curve. 

### 4.2. Pneumonia and Associated Environmental Factors

Regarding the incidence rate of pneumonia, we discovered that September, the rainy season in Thailand, had the highest prevalence, corresponding to the results of Tasci’s study that found a positive correlation between pneumonia and rain [17]. However, unlike influenza, the seasonality of pneumonia in Chiang Mai was unremarkable when compared to the reference month, with two peaks of high incidence recorded from August to November and January to March. Multiple NBR models revealed that temperature and its lag, relative humidity lag1, and PM_10_ were strongly correlated and had a linear relationship with the incidence rate of pneumonia. Similar to the results of the study conducted in Lampang, a province adjacent to Chiang Mai, it was discovered that relative humidity and PM_10_ were positively linked with the incidence of pneumonia [50]. In terms of pneumonia incidence forecasting based on air pollutants, neither the lag1 of PM_2.5_ nor the lag1 PM_10_ was significantly associated with the pneumonia incidence rate. A model with the composition of PM_2.5_ lag1 and temperature lag1 was shown to be significant at an alpha level of 0.1 (data not shown). From our results, the temperature and relative humidity data of the preceding month (lag1) were appropriate for use as potential predictors of pneumonia. According to the adjusted IRR, as shown in Table 3, temperature and PM_10_ had a greater impact on the influenza incidence than pneumonia. There was a negative correlation between temperature and the incidence of pneumonia, with the same direction of association reported for influenza. A lower temperature was associated with an increase in pneumonia hospitalizations [51,52]. According to Pirozzi’s study, the effects of PM_2.5_ were greatest during the colder months [53]; however, there was a lack of evidence regarding PM_2.5_ having a direct effect on the incidence of pneumonia, as the monthly concentration of PM_2.5_ was the highest during the summer and the NBR model did not demonstrate a significant association between PM_2.5_ and pneumonia. In terms of seasonality, however, the majority of the pneumonia cases in this study were reported during the rainy season rather than the cold season (Figure 3B). 

In addition to temperature, it has been observed that humidity and PM_10_ have positive relationships with severe pneumonia occurrence [52], which is consistent with our findings. The systematic review of short-term exposure to air pollution and hospital admission for pneumonia revealed that an increase of PM_10_ at every 10 μg/m^3^ was associated with a 0.4% (95%CI = 0.2, 0.6) increase in hospital admission or emergency room (ER) visits for pneumonia [54], which corresponded closely to the 0.2% (95%CI = 0.02, 0.42) increased incidence generated by our NBR model. A systematic review of PM exposure in children found that PM_10_ was associated with a 1.5% increase in pneumonia hospital admissions [55] as a result of their increased inhalation per body weight and immature immune systems, making them more susceptible to infections [56]. Given that PM_2.5_ has a smaller diameter and a higher influence on health than PM_10_, a short-term PM_2.5_ exposure study in Wuhan, China, revealed a maximum risk ratio (RR) of 1.20 for pneumonia outpatient visits [57]. In addition to the short-term exposure, a 12-month moving average PM_2.5_ exposure (per 10 μg/m^3^ increase) was associated with pneumonia mortality, with an RR of 1.45 in a large cohort study of long-term PM_2.5_ exposure in the United States [16]. In addition to viruses, bacteria and fungi are known to cause pneumonia as bioaerosols. With aerodynamic dimensions that are larger than viruses (in the range of micrometers), the average hit frequency was found to be greater in the PM_10_ samples than in the PM_2.5_ samples [58]. Less is known about the risk differences between genders, although evidence on the reproductive and birth outcomes points to pregnant women as a vulnerable group that has not been clearly identified [56]. For a better understanding of the disease, environmental factors, as well as demographic associations and age and gender distributions, should be examined in depth.

### 4.3. The Role of PMs in Respiratory Diseases in the Northern Region of Thailand

Annually, the northern region of Thailand is burdened by severe smog due to crop residue burning, which varies from season to season, mostly occurring during the dry season [59]. Compared to the yearly standards for PM_2.5_ and PM_10_ of 25 μg/m^3^ and 50 μg/m^3^, respectively (Table 1), it was determined that the concentrations surpassed the standard level throughout the months of December and May, which accounted for half the year. In addition to crop residue burning, drought and forest fires typically occur between December and April. The entire upper northern region of Thailand is affected by haze due to these activities [59,60,61]. In addition to being linked to the negative health effects of human biomass burning, smoke from wildfires, which contains numerous hazardous air pollutants, including PMs, is positively correlated with respiratory diseases, such as pneumonia [62,63,64]. As revealed by the NBR models, PMs had certain effects on the diseases. PM_2.5_ and PM_10_ are notorious for their adverse effects on human health. The primary mode of exposure to PMs is inhalation. Thus, the airways or respiratory tract have been a major focus of PMs research. PMs of smaller size have a stronger short-term influence on the respiratory system, particularly in vulnerable populations [20,65,66], due to the ability of the particle to pass through the filtration mechanism in the respiratory tracts and enter the lower tract. Compared to PM_2.5_ and PM_10_, PM_1_ has been found to be associated with a greater incidence of hospitalizations for childhood pneumonia, as reported by Wang’s study [65]. PMs induce oxidative stress and inflammation, stimulate innate and adaptive immunity, and other processes, which contribute to the onset and progression of respiratory illnesses [67,68]. There have been some studies investigating the association between daily PM_2.5_ exposure and hospital visits for respiratory diseases and a positive non-linear relationship between the variables has been found [19,20].

Due to the fact that the influenza, pneumonia, and environmental data used in our study were represented by monthly data, analyses of the association between the monthly incidence of the diseases and variable exposures, including lag1, were carried out instead of using daily data. However, Somayaji’s study involving a sensitivity analysis with weekly events, as opposed to daily events, revealed no significant differences [8]. We assumed that the monthly time-series study would have no impact on the aims of our investigation, given that the outcomes of this study were addressed within the context of seasonality. Compared to the number of cases received from the AESR that were used for analysis, the number of cases collected from the HDC PM_2.5_, which were air-pollution-related cases directly reported to the Bureau of Epidemiology, was lower (75% and 70% of the cases reported to the AESR for influenza and pneumonia, respectively). However, the cases reported to the HDC PM_2.5_ system were not a subset of those reported to the AESR, as the number of cases submitted to the HDC PM_2.5_ system, in some months, exceeded the number of cases reported to the AESR. On the website, no inclusion criteria for cases involving air pollution were made available to the public. 

Prior to the COVID-19 epidemic, people in Thailand avoided masks when experiencing respiratory symptoms. In recent years, as a result of the severity of PM_2.5_ concerns, which are most prevalent from February to April in the northern region, there has been an increase in the habit of wearing masks during excessive air pollution. Additionally, the appearance of COVID-19, which is an emerging respiratory disease, has altered the standards of personal protection, namely, the use of masks, and indirectly protects the public from air pollution problems. Regarding Li’s study [41] mentioned earlier, changes in respiratory illness incidences related to protective behaviors taken due to COVID-19 and the association with air pollutants were not explored in this study due to the fact that the study period was limited to 2020, which was the year in which the COVID-19 epidemic first began. Consequently, additional research is suggested, and the use of masks during the COVID-19 pandemic should be considered as a protective factor for further research into the seasonality of respiratory diseases and their associated factors.

### 4.4. Study Limitations

This study has a number of limitations. First, the primary limitation of the study was that data used for analysis were not collected directly from the monitoring of air-pollution-related diseases (the HDC PM_2.5_ database). Since mid-2020, air-pollution-related diseases have been subjected to nationwide health monitoring. Consequently, this limitation was apparently a gap that we wanted to focus on in the current study by using the AESR data to identify associations between the environmental factors and diseases of interest. In addition, all disease causes were included in the health data utilized for this analysis, regardless of the influenza virus type or pneumonia causes identified. Second, the climate data may not be representative of the entire areas for which diseases were reported. Due to the vast differences between the data received from the two weather stations, we decided to only include the data from the station in the Muang district, where the data more accurately represented the meteorological conditions of the living areas than the data from the station in the hilly region. Limitations in climate and air pollutant data led to limitations in environment-related health data analysis for the study area and other locations in the country, resulting in recommendations to policymakers regarding the future installation of additional monitoring stations so as to improve climate and air pollutant monitoring for health data forecasting. Third, due to the self-limiting characteristics of the diseases in mild cases, one constraint cited was the under-reporting of cases due to the difficulties in estimating true incidences. In 2020, there were approximately one-third fewer influenza cases than in the previous year, and the lowest number of pneumonia cases as compared to previous years. This is possibly because COVID-19 was brought to Thailand in 2020 and the number of infected cases has been steadily increasing. Our premise was that the suspected cases were predominantly screened for COVID-19 infections based on their symptoms. More cases were screened for COVID-19 than influenza or pneumonia, resulting in fewer influenza and pneumonia cases reported than in previous years. Fourth, we studied the impact of meteorological and air pollution conditions on influenza and pneumonia in this study. Consequently, we mainly observed environmental variables. To our knowledge, environmental factors are not the only factors contributing to influenza and pneumonia infections. Due to the communicable properties of the diseases, additional predisposing factors, particularly the pathogens’ characteristics, influence the occurrence of these diseases.

## 5. Conclusions

In this study, we analyzed the monthly environmental (meteorological and air pollutant) data to identify relationships with the respiratory diseases related to air pollution. The data indicated that influenza and pneumonia incidences peak during the cold and wet seasons, respectively. In terms of forecasting, the NBR models determined that PM_2.5_ and the temperature of the prior month (lag1) could be utilized to predict influenza incidence. In addition, the previous month’s temperature and relative humidity could be utilized to forecast pneumonia. This study will be useful for predicting the future dynamics of influenza and pneumonia incidences in the region due to air pollution and for allocating clinical and public health resources in response to environmental change scenarios.

## Figures and Tables

**Figure 1 tropicalmed-07-00341-f001:**
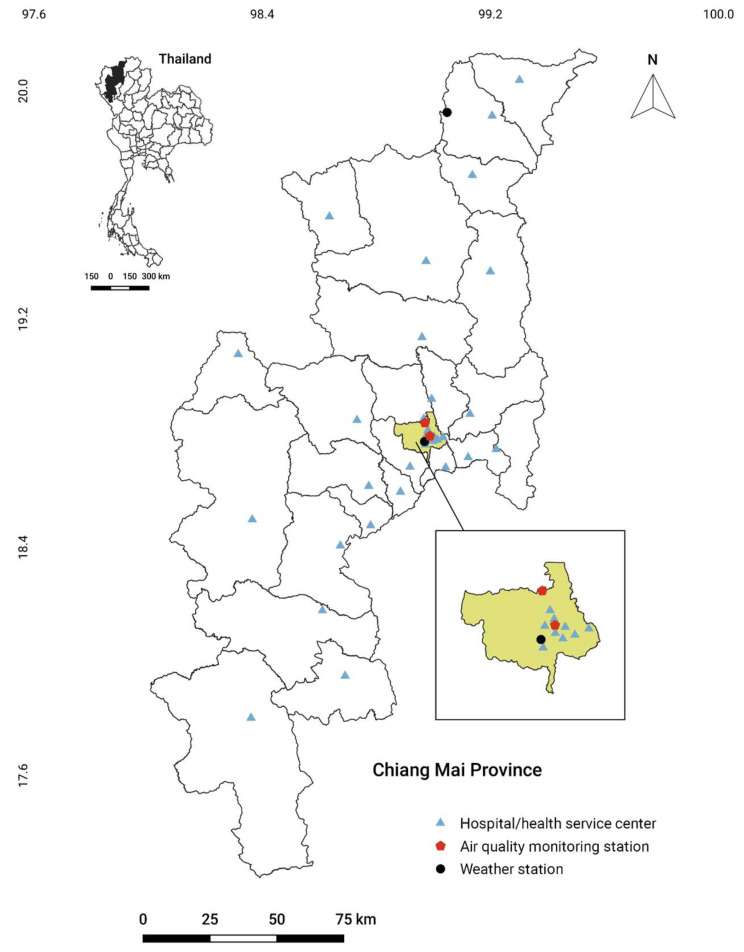
Map of Chiang Mai province with the location of weather stations, air quality monitoring stations, and district hospitals/health service centers in the health service network of air-pollution-related illness. Illustration of hospitals/health service centers located in Muang district is shown in an enlarged view. Details of hospitals/health service centers and coordinates are shown in Appendix A.

**Figure 2 tropicalmed-07-00341-f002:**
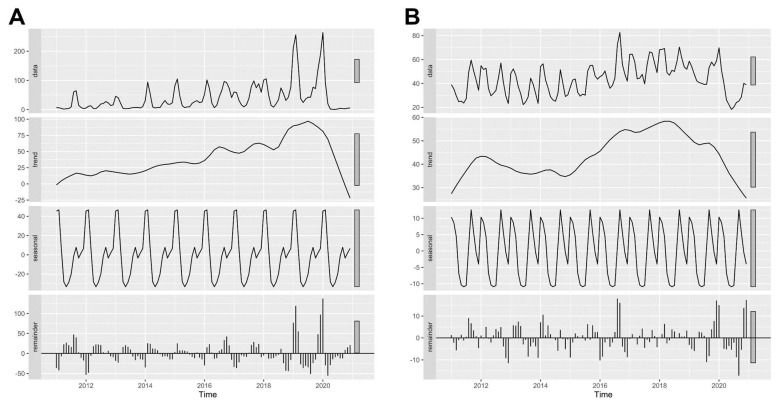
Seasonal-trend decomposition using loess (STL) plots of (**A**) influenza and (**B**) pneumonia in Chiang Mai from 2011 to 2020. A data column of the plot showed the original data of the monthly incidence rate, which were decomposed to trend, seasonal, and remainder plots.

**Figure 3 tropicalmed-07-00341-f003:**
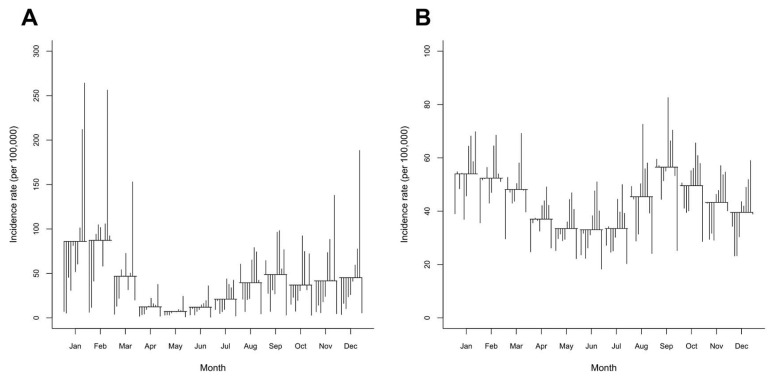
Seasonal cycle subseries (SCS) plots of the monthly incidence rates of (**A**) influenza and (**B**) pneumonia in Chiang Mai from 2011 to 2020. The SCS has a horizontal line for the average incidence rate of each month over the entire period, while each vertical line depicts the unique pattern for the same months in each year.

**Figure 4 tropicalmed-07-00341-f004:**
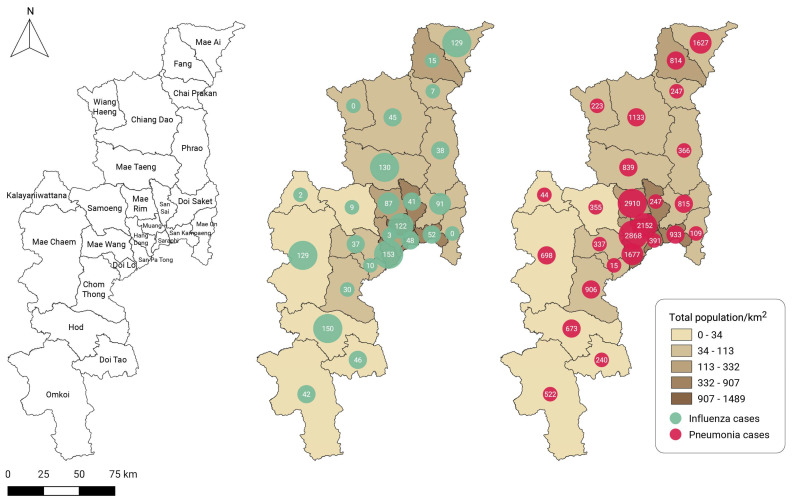
District administrative areas of Chiang Mai province (**left**). Cumulative case reports by district for (**middle**) influenza and (**right**) pneumonia in Chiang Mai obtained from the Bureau of Epidemiology’s PM_2.5_ health center data (HDC) database between July 2020 and June 2022.

**Figure 5 tropicalmed-07-00341-f005:**
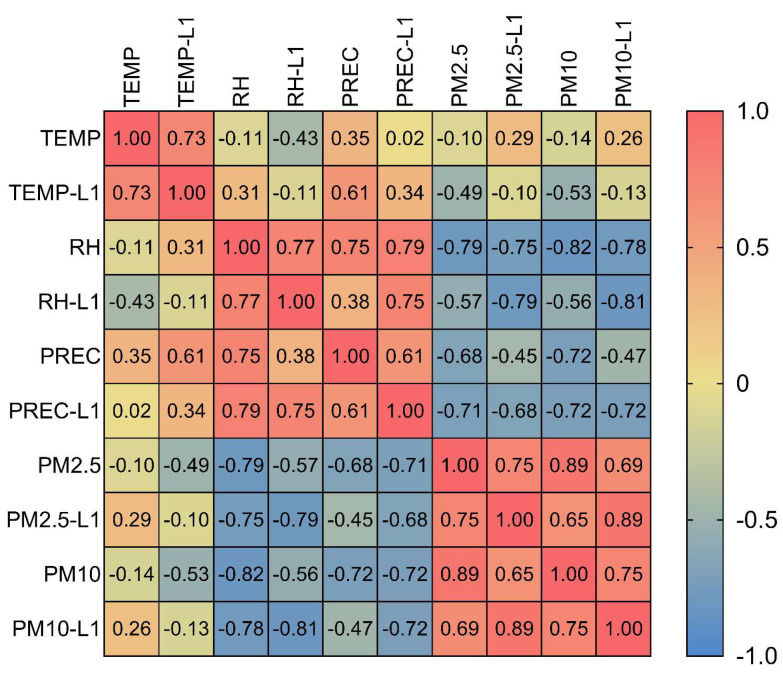
Spearman’s correlation matrix among environmental variables. TEMP, temperature; L1, lag1; RH, relative humidity; PREC, precipitation (rainfall).

**Table 1 tropicalmed-07-00341-t001:** Monthly average meteorological and air pollutant data in Chiang Mai from 2011 to 2020.

Season	Month	Temperature (°C) ^a^	Relative Humidity (%) ^a^	Total Precipitation (mm) ^a,b^	PM_2.5_ (µg/m^3^) ^c^	PM_10_ (µg/m^3^)
Dry, Cool	Jan	22.9	68.7	21.8	32.6	48.9
	Feb	25.1	58.6	7.8	49.5	68.6
Hot	Mar	28.0	53.5	13.5	80.9	101.7
	Apr	30.0	56.8	45.2	57.6	74.4
	May	29.6	67.8	172.4	25.1	38.8
Rainy	Jun	28.7	73.4	114.4	14.7	24.9
	Jul	28.0	76.5	154.9	14.6	25.1
	Aug	27.6	79.6	220.9	14.6	26.1
	Sep	27.8	78.5	189.4	14.2	26.4
	Oct	27.2	76.0	131.5	18.1	31.8
Dry, Cool	Nov	26.1	72.7	40.9	20.9	34.2
	Dec	23.3	71.1	9.4	28.6	44.3
	Average	27.0	69.4	93.5	22.0 ^d^	35.8 ^d^
	Standard [22]	-	-	-	50 ^e^/25 ^f^	120/50

^a^ Meteorological data used in this study were only retrieved from station#327501. ^b^ Total precipitation in Thailand only refers to total rainfall. ^c^ From January to April 2011, PM_2.5_ data were unavailable from both stations, and from January 2011 to June 2016, only data from station#36T could be retrieved. ^d^ As a result of the lognormal distribution of particulate matter (PM) concentration over ten years, the median concentrations of PM_2.5_ and PM_10_ were instead computed. Standard of PM defined by the Pollution Control Department of Thailand. ^e^ The 24-h average. ^f^ The annual average (µg/m^3^).

**Table 2 tropicalmed-07-00341-t002:** Univariate negative binomial regression (NBR) results of influenza and pneumonia incidence rates by month.

Month	Influenza IRR (95%CI)	Pneumonia IRR (95%CI)
Jan	12.01 (5.51, 26.19) *	1.64 (1.31, 2.04) *
Feb	12.18 (5.58, 26.57) *	1.59 (1.27, 1.98) *
Mar	6.53 (2.99, 14.25) *	1.46 (1.17, 1.82) *
Apr	1.73 (0.79, 3.78)	1.12 (0.90, 1.40)
May	Reference = 1	1.01 (0.81, 1.27)
Jun	1.68 (0.77, 3.67)	Reference = 1
Jul	2.93 (1.34, 6.39) *	1.01 (0.81, 1.27)
Aug	5.52 (2.53, 12.05) *	1.38 (1.10, 1.72) *
Sep	6.80 (3.12, 14.84) *	1.71 (1.37, 2.14) *
Oct	5.16 (2.36, 11.25) *	1.50 (1.20, 1.87) *
Nov	5.80 (2.66, 12.66) *	1.31 (1.05, 1.64) *
Dec	6.31 (2.89, 13.76) *	1.20 (0.96, 1.50)

IRR, incidence rate ratio; CI, confidence interval. * Statistically significant at *p* < 0.05.

**Table 3 tropicalmed-07-00341-t003:** Final NBR models with associated environmental risk factors for respiratory disease incidences in Chiang Mai from 2011 to 2020.

Model	Input Predictors	Model Equation	Adjusted IRR	95%CI	*p*-Value
Influenza final NBR models					
1	PM_2.5_ lag1,	lnIF =−8.10 + 0.03(PM_2.5_l1) − 0.0004(PM_2.5_l1sq) *	1.027	0.989, 1.067	0.164
	PM_2.5_ lag1_sq		1.000	0.999, 1.000	0.023
2	PM_10_	lnIF = −8.41 + 0.01(PM_10_) **	1.012	1.004, 1.021	0.004
3	Temperature lag1	lnIF = −1.29 − 0.25(Templ1) ***	0.781	0.711, 0.858	<0.001
Pneumonia final NBR models					
4	PM_10_	lnPN = −7.83 + 0.002(PM_10_) *	1.002	1.000, 1.004	0.033
5	Temperature	lnPN = −6.84 − 0.03(Temp) *	0.967	0.940, 0.995	0.021
6	Temperature lag1,	lnPN = −6.87 − 0.05(Templ1) ** + 0.01(Rhl1) **	0.951	0.919, 0.983	0.003
	Relative humidity lag1		1.007	1.002, 1.012	0.003

lag1, data from the preceding month; lag1_sq, lag1 with quadratic term; IF, influenza; PN, pneumonia; IRR, incidence rate ratio; CI, confidence interval. Predictor’s significance level: ‘*’, 0.05; ‘**’, 0.01; ‘***’, 0.001.

## Data Availability

The data presented in this study are available on request from the corresponding author. The data are not publicly available due to the institutional privacy policy.

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
