# Peer review of "Air Pollution-Related Respiratory Diseases and Associated Environmental Factors in Chiang Mai, Thailand, in 2011–2020"

_tropicalmed, 2022, doi:10.3390/tropicalmed7110341_

Round 1

Reviewer 1 Report

In this study, author analyzed the relationship between environmental factors (meteorological and air polluioan) and respiratory disease in Thainland. The manuscript is well wrtitten and methods described in detail. This study is very important because it provides evidence on the effects of air pollution in a region that has been little studied and that we know that has high levels of air pollution. In addition, it is very important to support local public health and environment policies.

Authors should only revise introduction because it is too long.

Author Response

Dear Reviewer,

We appreciate you dedicating your time for reading and commenting on our work. Now, we complete responding all of your issues/queries. Please see an attachment for our responses.

Best Regards,

Reviewer 2 Report

Air pollution is a known contributor to respiratory disease. Both air pollution and respiratory disease are seasonal, and understanding the association between air pollution and respiratory disease in a region may guide policy to reduce exposure and protect health. The purpose of this study was to evaluate the association between air pollution and respiratory disease in Chiang Mai, Thailand, taking into consideration the seasonality of both of these variables. While it is certainly appreciated that this information is important to policy making in this region, it is unclear what additional information it contributes to the knowledge base. Overall, the length and lack of clarity/focus of the manuscript dilutes the primary purpose/outcome and makes it difficult for the reader to interpret. Further, it is unclear, besides the availability of the data, what additional insight this study contributes to the general knowledge base outside of being informative for motivating policy change in Chiang Mai itself. I have the additional major and minor comments to offer:

Major Comments:

1.     The authors indicate that Chiang Mai considered to be the most polluted city on earth during the hot season. This feels factually incorrect. Additionally, the reference for this is not appropriate.

2.     There are major limitations in exposure classification given that only one meteorologic monitoring station and two air pollution monitoring stations were used. If data from the whole region was used for respiratory outcomes, it seems that data from both environmental monitoring stations should be included in the study. Lack of spatial variability in environmental monitoring may particularly be a problem given the data provided on the variability of respiratory disease by district.

3.     The manuscript feels unnecessarily long, with detail not pertinent to the primary purpose of the study (for example, line 137 describing the four types of influenza virus). The writing feels more like that of a graduate thesis than a peer-reviewed manuscript. The volume of extraneous information makes it a bit difficult to comprehend the key points of the manuscript (such as the methods).

4.     The introduction does not fully justify the need for this particular study. Particulate matter, for example, has already been established as a risk factor for respiratory disease. What additional gap does this study fill?

No minor comments.

Author Response

(The authors gave the same response as above.)

Reviewer 3 Report

The authors have provided a detailed analysis on the association of environmental variables with the incidence of influenza and pneumonia in Chiang Mai, Northern Thailand. This is a region where high particle pollution due to annual intense biomass burnings at the end of the dry season. The environment variables include temperature, humidity, rain, PM2.5 and PM10. As there is no data for other respiratory diseases, only available influenza and pneumonia data is used. Both of these diseases are caused by virus or bacteria. The authors used negative binomial regression (NBR) for the analysis of the association of environmental variables and these two diseases by breaking down the data to trend, seasonal and residual components. The results are expected with temperature and particles (PM2.5 or PM10) having strong influence on the rate of disease incidences compared to relative humidity or rain.

The authors should discuss more on the link of virus attached to the particles entering the body into the respiratory system. There are a number of recent researches on this viral attachment to particles especially in relation to covid-19 virus. 

Overall I recommend the manuscript to be accepted for publication with minor revision

Some specific comments are listed below

(1) Line 87: Change "to observe that Bangkok’s" to "to find that Bangkok’s"

(2) Line 201: "STATA command". Should specify what software package used for the analysis ?. Should change this to "the 'nbrg' command in STATA software package".

(3) Line 215: Change "due to showing" to "for"

(4) Line 224: Spearman not Spearmen

(5) Line 291: Delete "health". And change "directly" to "which are directly"

(6) Line 319: "This predicted" should be "This model predicted"

Author Response

(The authors gave the same response as above.)

Round 2

Reviewer 2 Report

The authors have sufficiently addressed reviewer concerns.